# rDNA Copy Number Variation and Methylation in Human and Mouse Sperm

**DOI:** 10.3390/ijms26094197

**Published:** 2025-04-28

**Authors:** Ramya Potabattula, Marcus Dittrich, Thomas Hahn, Martin Schorsch, Grazyna Ewa Ptak, Thomas Haaf

**Affiliations:** 1Institute of Human Genetics, Julius Maximilians University, 97074 Würzburg, Germany; ramya.potabattula@uni-wuerzburg.de (R.P.); marcus.dittrich@uni-wuerzburg.de (M.D.); 2Department of Bioinformatics, Julius Maximilians University, 97074 Würzburg, Germany; 3Fertility Center, 65189 Wiesbaden, Germany; 4Malopolska Centre of Biotechnology (MCB), Jagiellonian University, 30-387 Krakow, Poland; g.ptak@uj.edu.pl

**Keywords:** absolute rDNA copy number, active rDNA copy number, aging human and mouse germline, deep bisulfite sequencing, droplet digital PCR, sperm rDNA methylation

## Abstract

In this study, droplet digital PCR and deep bisulfite sequencing were used to study the absolute and active rDNA copy number (CN) and the effect of paternal age on human and mouse sperm. The absolute CN ranged from 98 to 404 (219 ± 47) in human and from 98 to 177 (133 ± 14) in mouse sperm. Methylation of the human upstream control element/core promoter (UCE/CP) region and the 5′ external transcribed spacer, as well as that of the mouse CP, the spacer promoter, and 28S rDNA, significantly increased with donor age and absolute CN. Overall, rDNA hypomethylation was much more pronounced in mouse sperm, with 101.7 ± 11.4 copies showing a completely (0%) unmethylated and 11.3 ± 2.8 (8.5%) a slightly methylated (1–10%) CP region, compared to humans with 25.7 ± 9.5 (12%) completely unmethylated and 83.0 ± 19.8 slightly methylated UCE/CP regions. Although the absolute CN was much higher in human sperm, the number of copies with a hypomethylated (0–10%) promoter was comparable in humans (108.7 ± 28.3) and mice (113.0 ± 12.2). However, in mice, the majority (77%) of all copies were completely unmethylated, whereas in humans a high percentage (38%) showed one or two single CpG methylation errors. These different germline methylation dynamics may be due to species differences in reproductive strategies and lifespan. Complete demethylation of the sperm rDNA promoter in mice may be essential for embryonic genome activation, which already occurs at the 2-cell stage in mice and at the 4–8-cell stage in humans. The paternal age effect has been conserved between humans and mice with some notable differences. In humans, the number of hypomethylated (0–10%) copies decreased with age, whereas in mice only the completely unmethylated copies decreased with age. The number of methylated rDNA copies (>1% in mice and >10% in humans) significantly increased with age.

## 1. Introduction

The sperm epigenome is the product of male germline reprogramming, which is affected by intrinsic (i.e., cis-acting genetic variants) and extrinsic (i.e., male infertility and paternal age) factors [1,2]. In humans and other mammalian species, sperm methylation patterns display a bimodal distribution [3,4,5,6]. In human and primate sperm, most (75–85%) protein-coding genes are unmethylated (<5%) over their entire promoter region, and approximately 10% are fully (>95%) methylated [7]. It is generally assumed that sperm methylation patterns affect the transcriptional activation of genes in embryo development [7,8,9,10] and in long-term disease susceptibility later in life: “How well one builds an organism makes a great deal of difference on how long it lasts and how well it functions” [11].

Human ribosomal DNA (rDNA) is represented in the diploid human (blood) genome by several hundred (mean ± SD 469 ± 107; median 469; range 243–895) rDNA transcription units (TUs), which are tandemly arrayed in the nucleolus organizer regions of the acrocentric short arms [12]. Each rDNA TU contains more than 1500 CpG sites whose methylation is involved in epigenetic rDNA silencing [13,14,15]. The number of presumably active human rDNA TUs with a hypomethylated (0–10%) promoter region is considerably smaller (182 ± 35; median 180; range 94–277) and independent of absolute copy number (CN). Thus, promoter methylation compensates to some extent for the enormous interindividual CN variation.

The rRNAs along with 70–80 ribosomal proteins form the two subunits of the ribosome, the cellular machinery for messenger RNA translation and protein synthesis [16]. Ribosome biogenesis, which occurs in the nucleolus, is closely inter-related with cell metabolism, growth, proliferation, and the maintenance of homeostasis. rDNA methylation reflects changes in rDNA transcription, ribosome biogenesis, and nucleolar biology during development, aging, and in many age-related conditions, including cancer, metabolic, and cardiovascular disorders [17,18]. A series of elegant experiments in nematodes, fruit flies, and mice [19], as well as in Hutchinson–Gilford progeria syndrome [20], suggested that nucleolar size is shrinking during the aging process. Small nucleoli and decreased rRNA expression are associated with longevity. Vice versa, nucleolar expansion, increased ribosome biogenesis, and protein translation are associated with premature aging. A positive correlation between rDNA methylation and age has been observed in different rodent and human somatic tissues [21,22,23] as well as in germ cells [2,24]. Deep bisulfite sequencing (DBS) of individual rDNA TUs revealed that the aging of somatic tissues is associated with a loss of hypomethylated (0–10%) active rDNA copies and, by extrapolation, rRNA dosage, whereas the absolute CN remained unchanged [12].

Embryonic genome activation (EGA) is essential for the establishment of totipotency in an early embryo [25]. In mice, minor EGA is already initiated in the late one-cell stage, followed by major EGA in the two-cell stage [26]. Transcription of rDNA genes starts at the end of the two-cell stage and is required for further development. Experimental inhibition of rDNA transcription causes developmental delay and arrest [27,28,29]. In humans, EGA is initiated at least two days after fertilization at the 4–8-cell stage [30]. Considering the impact of ribosome biogenesis on essentially every cellular process, in particular the mass production of proteins during early development, it seems plausible to assume that hypomethylation of the paternal rDNA genes that are transmitted via the sperm into the zygote affects their transcription after/during EGA. rDNA transcription plays an important role in nucleolar organization and higher-order functional genome architecture in an early embryo [27,28].

It is well known that absolute rDNA CN in somatic tissues can considerably vary between individuals and also between species. We assume that redundant copies are inactivated by rDNA methylation. Consequently, active CN should be less variable. Recently, we developed a combination of droplet digital PCR (ddPCR) and DBS, which allows us to count not only the absolute rDNA CN but also the number of presumably active copies with a hypomethylated promoter region with an unprecedented accuracy. To find out whether haploid germ cells exhibit the same CN variation and age-dependent gain of methylation as somatic cells (blood) [12], here, we determined the number of rDNA TUs with 0%, 10%, 20%, etc., methylation in human and mouse sperm. Despite enormous species differences in absolute rDNA CN, the number of hypomethylated copies and the paternal age effect were comparable between human and mouse sperm. An evolutionarily conserved rDNA methylation clock [2,14] appears to operate in similar ways in the mammalian soma and germ line. So far little is known about the functional consequences of rDNA CN variation, which has been largely neglected in reproduction.

## 2. Results

### 2.1. Human Sperm Samples

Using ddPCR, the absolute rDNA CN in normozoospermic sperm samples of 94 donors (from 29 to 72 years) ranged from 98 to 404 (219 ± 47; median 214, IQR 62). There was no significant correlation (Spearman *ρ* = 0.09; *p* = 0.39) between absolute CN and age (Appendix A). Using DBS, the upstream control element (UCE) and core promoter (CP) region as well as the 5′ external transcribed spacer (ETS) showed highly significant correlations between methylation and age (UCE/CP *ρ* = 0.68; *p* < 0.0001 and ETS *ρ* = 0.60, *p* < 0.0001) (Appendix A). In addition, methylation was strongly correlated with absolute CN (UCE/CP *ρ* = 0.38; *p* < 0.0001 and ETS *ρ* = 0.34, *p* < 0.0001) (Appendix A). Mean UCE/CP methylation (12.0 ± 2.9%; median 11% range 7.1–20.5%) was higher than that of the ETS (8.2 ± 1.7%; median 8% range 5.0–14.5%).

To study the age-related methylation changes in more detail, we classified the CN in different methylation bins (0%, 1–10%, 11–20%, 21–30%, 31–40%, 41–50%, and 51–100%). Human sperm were endowed with 25.7 ± 9.5 (median 26, range 7–53) rDNA copies with a completely unmethylated (0%) UCE/CP and 83.0 ± 19.8 (median 82.6, range 37–138) copies with a slightly methylated (1–10%) CP (Appendix A). A similar distribution was observed for the ETS with 26.7 ± 9.3 (median 25, range 10–53) unmethylated and 116.3 ± 26.3 (median 116, range 60–211) slightly methylated copies. The remaining TUs (110.7 ± 39.9 for the UCE/CP and 76.4 ± 30.4 for the ETS) showed more than 10% methylation.

When plotting the number of TUs within every bin against age, the number of completely unmethylated TUs (UCE/CP *ρ* = −0.64, *p* < 0.0001; ETS *ρ* = −0.41, *p* < 0.0001) and slightly methylated TUs (UCE/CP *ρ* = −0.50, *p* < 0.0001; ETS *ρ* = −0.29, *p* = 0.005) was significantly decreasing with donor age (Figure 1). In contrast, the number of TUs with 11–20% (UCE/CP *ρ* = 0.33, *p* = 0.001; ETS *ρ* = 0.43, *p* < 0.0001), 21–30% (UCE/CP *ρ* = 0.61, *p* < 0.0001; ETS *ρ* = 0.59, *p* < 0.0001), 31–40% (UCE/CP *ρ* = 0.63, *p* < 0.0001; ETS *ρ* = 0.52, *p* < 0.0001), 41–50% (UCE/CP *ρ* = 0.53, *p* < 0.0001; ETS *ρ* = 0.35, *p* = 0.0006), and 51–100% (UCE/CP *ρ* = 0.41, *p* < 0.0001; ETS *ρ* = 0.19, *p* = 0.07) was significantly increasing with age.

The core promoter is the most important regulatory element for rDNA transcription. Promoter (hyper)methylation leads to rDNA silencing. Consistent with our previous study on blood rDNA methylation [12], we classified TU copies with 0–10% mean methylation as presumably active (after fertilization) and copies with >10% methylation as hypermethylated and inactive. The number of hypomethylated (0–10%) TUs was significantly decreasing (UCE/CP *ρ* = −0.56, *p* < 0.0001; ETS *ρ* = −0.35, *p* = 0.0005) with age, whereas the number of hypermethylated (11–100%) TUs was significantly increasing (UCE/CP *ρ* = 0.49, *p* < 0.0001; ETS *ρ* = 0.49, *p* < 0.0001) (Figure 2A). Our results show that 108.7± 28.3 (median 108, range 43–191) rDNA copies in human sperm are hypomethylated (0–10%) and presumably active after fertilization; 110.7± 39.9 (median 104, range 38–238) copies are hypermethylated (>10%) and inactive.

For both the UCE/CP (Figure 3A) and the ETS (Figure 3B), the number of TUs in all methylation bins was positively correlated with absolute CN: 0% (UCE/CP *ρ* = 0.20, *p* = 0.05; ETS *ρ* = 0.28, *p* = 0.007), 1–10% (UCE/CP *ρ* = 0.60, *p* < 0.0001; ETS *ρ* = 0.81, *p* < 0.0001), 11–20% (UCE/CP *ρ* = 0.87, *p* < 0.0001; ETS *ρ* = 0.72, *p* < 0.0001), 21–30% (UCE/CP *ρ* = 0.61, *p* < 0.0001; ETS *ρ* = 0.56, *p* < 0.0001), 31–40% (UCE/CP *ρ* = 0.58, *p* < 0.0001; ETS *ρ* = 0.55, *p* < 0.0001), 41–50% (UCE/CP *ρ* = 0.59, *p* < 0.0001; ETS *ρ* = 0.44, *p* < 0.0001), and 51–100% (UCE/CP *ρ* = 0.54, *p* < 0.0001; ETS *ρ* = 0.33, *p* < 0.0001). Both hypomethylated (0–10%) rDNA TUs (UCE/CP *ρ* = 0.48, *p* < 0.0001; ETS *ρ* = 0.72, *p* < 0.0001) and hypermethylated (11–100%) TUs (UCE/CP *ρ* = 0.78, *p* < 0.0001; ETS *ρ* = 0.72, *p* < 0.0001) showed a strong positive correlation with absolute CN (Figure 2B).

### 2.2. Mouse Sperm Samples

To test whether the effect of age and absolute CN on sperm rDNA methylation has been conserved during mammalian evolution, we analyzed the CN in 173 3–16-month-old C57BL/6 mice. The absolute rDNA CN in mouse sperm ranged from 98 to 177 (133 ± 14; median 133, IQR 20) and was considerably smaller than in humans. Similar to humans, absolute CN did not change with age (Spearman *ρ* = 0.04; *p* = 0.65) (Appendix A). Using DBS, we analyzed the methylation of the core promoter (CP), the spacer promoter (SP), and 28S rDNA. The mean methylation of all three regions was positively correlated with age (CP *ρ* = 0.31; *p* < 0.0001; SP *ρ* = 0.39; *p* < 0.0001; 28S *ρ* = 0.43, *p* < 0.0001) (Appendix A). Methylation of the CP (6.4 ± 2.7%; median 5.9%, range 1.7–21.2%) and SP (5.6 ± 2.5%; median 5.2%, 1.6–15.2%) was lower than that of the 28S rDNA region (13.1 ± 3.9%; median 13.1%, range 6.0–23.9%). In addition, there also was a positive correlation between regional methylation and absolute CN (CP *ρ* = 0.22; *p* = 0.003; SP *ρ* = 0.18; *p* = 0.02; 28S *ρ* = 0.29, *p* < 0.0001) (Appendix A). In fact, most rDNA copies were completely unmethylated (CP 101.7 ± 11.4; SP 93.4 ± 11.7; 28S 70.0 ± 10.3) in mouse sperm (Appendix A).

Next, we classified the CN in different methylation bins. For all three target regions, the number of completely unmethylated (0%) TUs decreased with age (CP *ρ* = −0.28, *p* = 0.0002; SP *ρ* = −0.44, *p* < 0.0001; 28S *ρ* = −0.63, *p* < 0.0001). In all other methylation bins, CN increased with age (Appendix A): 1–10% (CP *ρ* = 0.57, *p* < 0.0001; SP *ρ* = 0.58, *p* < 0.0001; 28S *ρ* = 0.43, *p* < 0.0001), 11–20% (CP *ρ* = 0.51, *p* < 0.0001; SP *ρ* = 0.71; *p* < 0.0001; 28S *ρ* = 0.67, *p* < 0.0001), 21–30% (CP *ρ* = 0.57, *p* < 0.0001; SP *ρ* = 0.44, *p* < 0.0001; 28S *ρ* = 0.68, *p* < 0.0001), 31–40% (CP *ρ* = 0.27, *p* = 0.0004; SP *ρ* = 0.35, *p* < 0.0001; 28S rDNA *ρ* = 0.40, *p* < 0.0001), 41–50% (CP *ρ* = 0.21, *p* = 0.004; SP *ρ* = 0.31, *p* < 0.0001; 28S *ρ* = 0.34, *p* < 0.0001), and 51–100% (CP *ρ* = 0.18, *p* = 0.018; SP *ρ* = 0.26, *p* = 0.0005; 28S *ρ* = 0.29, *p* = 0.0001). In mouse sperm, only the completely unmethylated alleles were negatively correlated with age (see above), whereas the methylated (1–100%) alleles showed a strong positive correlation (CP *ρ* = 0.45, *p* < 0.0001; SP *ρ* = 0.59, *p* < 0.0001; 28S *ρ* = 0.58, *p* < 0.0001) (Figure 4A).

In the mouse sperm, 101.7 ± 11.4 (median 101, range 76–131) copies had a completely unmethylated CP and 11.3 ± 2.8 (median 11, range 6–20) copies were slightly (1–10%) methylated. Overall, 113.2 ± 12.2 (median 113, range 85–145) rDNA promoter regions were hypomethylated (0–10%) and only 20.4 ± 6.4 (median 21.5. range 7–48) were hypermethylated (11–100%) in the mouse sperm.

Similar to humans, rDNA methylation increased with absolute CN in all methylation bins (Appendix A): 0% (CP *ρ* = 0.80, *p* < 0.0001; SP *ρ* = 0.70, *p* < 0.0001; 28S *ρ* = 0.49, *p* < 0.0001), 1–10% (CP *ρ* = 0.57, *p* < 0.0001; SP *ρ* = 0.67, *p* < 0.0001; 28S *ρ* = 0.62, *p* < 0.0001), 11–20% (CP *ρ* = 0.65, *p* < 0.0001; SP *ρ* = 0.48; *p* < 0.0001; 28S *ρ* = 0.57, *p* < 0.0001), 21–30% (CP *ρ* = 0.59, *p* < 0.0001; SP *ρ* = 0.39, *p* < 0.0001; 28S *ρ* = 0.43, *p* < 0.0001), 31–40% (CP *ρ* = 0.53, *p* = 0.0004; SP *ρ* = 0.37, *p* < 0.0001; 28S rDNA *ρ* = 0.61, *p* < 0.0001), 41–50% (CP *ρ* = 0.50, *p* < 0.0001; SP *ρ* = 0.33, *p* < 0.0001; 28S *ρ* = 0.55, *p* < 0.0001), and 51–100% (CP *ρ* = 0.32, *p* < 0.0001; SP *ρ* = 0.29, *p* = 0.0001; 28S *ρ* = 0.44, *p* < 0.0001). Consequently, both the number of completely unmethylated copies (see above) and the number of methylated (1–100%) copies (CP *ρ* = 0.80, *p* < 0.0001; SP *ρ* = 0.56, *p* < 0.0001; 28S *ρ* = 0.67, *p* < 0.0001) were positively correlated with absolute CN (Figure 4B).

## 3. Discussion

### 3.1. Absolute rDNA CN and Methylation in the Mammalian Germline

The absolute rDNA CN in human sperm (219 ± 47; median 214, range 98–404) was approximately half of that in blood (469 ± 107; median 469, range 243–895) [12]. This is consistent with the haploidy of the sperm genome. Promoter (UCE/CP) methylation in human sperm (12.0 ± 2.9%) was considerably lower than in blood (25.9 ± 9.4%), whereas the ETS methylation levels (8.2 ± 1.7% in sperm and 10.5 ± 2.9% in blood) were comparable. In the mouse model, it has been shown previously that rDNA promoter methylation in sperm, oocytes, and early embryos is low compared to somatic tissues [31]. The absolute rDNA CN in mouse sperm (133 ± 14; median 133, range 98–177) was much smaller than that in human sperm. Moreover, hypomethylation of the CP (6.4 ± 2.7%) and SP (5.6 ± 2.5%) in mouse sperm was more pronounced than in the human promoter region. The number of copies with a completely unmethylated (0%) core promoter was much higher in mouse (101.7 ± 11.4) than in human sperm (25.7 ± 11.4). The majority (83.0 ± 19.8) of the UCE/CP regions in human sperm were methylated at low levels (1–10%). This may be associated with the longer lifespan and higher number of spermatogonial cell divisions in humans, during which single CpG methylation errors may occur. Evidently there are between-species differences in the absolute rDNA CN and methylation reprogramming in the male germline.

### 3.2. rDNA Promoter Methylation Differences in the Human and Mouse Germline

Although the absolute CN in human sperm (219 ± 47; median 214, range 98–404) was much higher than that in mouse sperm (133 ± 14; median 133, range 98–177), the number of copies with a hypomethylated (0–10%) promoter was comparable in both species (108.7 ± 28.3 in humans and 113.0 ± 12.2 in mice). In humans, 50% (108.7 of 219) of the rDNA promoter regions were hypomethylated (0–10%), whereas in mice, 77% (101.7 of 133) were completely unmethylated and another 8.5% methylated at low levels (1–10%). Interestingly, in both species the lowest observed absolute CN was 98. The lowest number of hypomethylated copies in humans was 43, and the lowest number of completely unmethylated copies in the mouse was 78. Collectively, these results suggest that an absolute CN of around 100 and an active CN of 40 to 80 rDNA TUs in sperm are required to ensure adequate rDNA activity in early embryos.

Despite the smaller absolute CN, the chances that a completely unmethylated (0%) or hypomethylated (0–10%) sperm fertilizes an egg are much bigger in mice (77% and >85%, respectively) than in humans (12% and 50%). This promotes the idea that there is strong evolutionary pressure on the mouse germline for complete demethylation of the rDNA core promoter. Human sperm may be more vulnerable to the accumulation of single CpG methylation errors. This may be compensated for by a higher absolute CN in humans, ensuring that the number of hypomethylated promoters does not fall below a critical threshold.

These species differences in absolute CN and methylation dynamics in the germline may be associated with the fact that EGA occurs at the mouse 2-cell stage but at the 4–8-cell stage in humans. Moreover, the fertility rate in mice is much higher than that in humans. Mice become sexually mature at 35–50 days, produce a litter of 8–10 pups, and have a reproductive span of two years. Humans become sexually mature at 13 years of age, give birth to one offspring at a time, and have a reproductive span of 40 years [32].

### 3.3. The Paternal Age Effect on Sperm rDNA Methylation in Humans and Mice

In both the human somatic tissue [12] and sperm, the number of completely unmethylated (0%) and slightly methylated (1–10%) rDNA TUs decreased with age, whereas the number of copies in all bins with >10% methylation increased. This paternal age effect on rDNA methylation was also observed in the mice, however, with one notable difference. In the mouse sperm, 77% (on average 101.7 of 133) of the rDNA TUs had a completely unmethylated CP, and 8.5% (11.3 of 219) were slightly methylated (1–10%). In humans, only 12% (25.7 of 219) of the UCE/CP regions were completely unmethylated, and 38% (83 of 219) were slightly methylated. In the mouse model, only the completely unmethylated rDNA TUs decreased with age, whereas copies with 1–10% or higher methylation increased. In this light, it seems plausible to assume that completely unmethylated paternal rDNA copies may be essential for mouse embryogenesis, whereas in humans, single CpG methylation errors are at least to some extent tolerated.

In both humans and mice, the number of hypomethylated and hypermethylated regions increases with absolute CN. This implies that donors with a high absolute CN also have more copies with a completely unmethylated or slightly methylated promoter region, which is generally thought to facilitate rDNA transcription in early embryos. Thus, from a reproductive point view, a high absolute CN may be advantageous.

### 3.4. Limitations

Methylation of the core promoter region plays a crucial role in switching an rDNA gene from active to inactive [13,15]. Because it is usually the density of methylated CpGs in a cis-regulatory region rather than individual CpGs that turns a gene “on” or “off” [33], methylation of 1–2 of 25 (4% and 8% regional methylation) CpGs in the human CP/UCE and 1 of 17 (6%) CpGs in the mouse CP were considered as single CpG methylation errors without functional consequences. Circumstantial evidence [12] suggests that rDNA TUs with 0–10% promoter methylation are active, and copies with >10% methylation are prone to epigenetic silencing. However, in this context, it is important to note that we do not really know whether and to what extent unmethylated (0%) or slightly (1–10%) methylated rDNA TUs are transcribed. It is difficult to prove by functional experiments, which methylated CpG density is required for epigenetic silencing, and there may not be a sharp threshold anyway. Moreover, there may be species differences. Our data suggest that slightly (1–10%) methylated rDNA copies are active in humans but not in mice.

Here, we compared the absolute and active rDNA CN in human and mouse sperm. Overall, the paternal age effect on rDNA methylation has been conserved, leading to a decrease in presumably active sperm rDNA copies. This may contribute to the effect of paternal age on offspring development. However, this is largely conjectural, as so far there is no evidence of a functional outcome in the post-fertilization early embryo. Because of species differences in reproduction, embryogenesis, and lifespan, it is difficult to extrapolate from the mouse model to humans and vice versa. As outlined above, “when it comes to studying ageing …, mice are not just small humans” [33].

## 4. Materials and Methods

### 4.1. Study Samples

The 94 human sperm samples were excess materials from assisted reproduction. The left-over swim-up sperm fractions after IVF/ICSI were collected at the Fertility Center Wiesbaden, pseudonymized, and frozen at −80 °C until further use. Donor age ranged from 29 to 72 years (39.3 ± 5.9; median 38.5) and BMI from 18.9 to 41.8 kg/m^2^ (25.6 ± 2.9; median 25.1). Sperm concentration was 84.5 ± 45.8 millions/mL, progressive motility 55.8 ± 13.8%, and normal morphology 7.2 ± 3.0%. Although the sperm donors were attending a fertility center, they had normal semen parameters according to the 5th edition of the WHO laboratory manual [34], which is generally considered as an indicator for male fertility potential. After thawing, the swim-up sperm samples were purified further by Silane-coated density gradients PureSperm 80 and 40 (Nidacon, Mölndal, Sweden).

Mouse (*Mus musculus*) sperm samples (N = 173) were isolated from 3–16-month-old fertile C57BL/6 mice after cervical dislocation. The vas deferens and caudal epididymis were dissected and placed separately into 500 µL GMOPS with 10 mg/mL human serum albumin at 37 °C. After repeated washing with 1× PBS, the final fraction was resuspended in 500 µL PBS and stored at −80 °C. All experimental procedures were conducted according to the guidelines of the European Community Regulations and conformed to the Polish Governmental Act for Animal Care. Animal procedures were approved by the II Local Ethics Commission of Krakow (123/2018, 318/2018, and 329/2021).

For DNA isolation, the purified sperm cells were resuspended in 300 µL buffer (5 mL of 5 M NaCl, 5 mL of 1 M Tris-HCl; pH 8, 5 mL of 10% SDS; pH 7.2, 1 mL of 0.5 M EDTA; pH 8, 1 mL of 100% β-mercaptoethanol, and 33 mL H_2_O) and 100 µL (20 mg/mL; 600 mAU/mL) proteinase K (Qiagen, Hilden, Germany) and incubated for 2 h at 56 °C. Sperm DNA was isolated using a DNeasy Blood and Tissue kit (Qiagen). DNA concentration and purity were measured by a NanoDrop 2000c spectrophotometer (Thermo Scientific, Waltham, MA, USA). Bisulfite conversion of DNA was performed using an EpiTect Fast 96 Bisulfite kit (Qiagen), and the converted DNA was stored at −20 °C until further use.

### 4.2. Droplet Digital PCR

ddPCR primers (Appendix A) for 28S rDNA, the human TATA-box binding protein (*TBP*) gene (internal control), and the mouse glyceraldehyde-3-phosphate dehydrogenase 1 (*Gapdh1*) gene (internal control) were taken from the literature [35]. To evaluate the CN of 28S rDNA, 1 ng of genomic DNA (per sample) was added to a mixture containing 10 µL of ddPCR Supermix for Probes (without dUTP, 2× concentrated), 1 µL each of 20× FAM (reference gene label) and 20× HEX (rDNA label) assay mixes, 0.2 µL of Hae III restriction endonuclease (10,000 units/mL; New England Biolabs, Frankfurt, Germany), and 6.8 µL of H_2_O. Probes labeled with distinct fluorophores for rDNA and the reference gene allowed precise quantification of both loci within a single duplex ddPCR reaction. Furthermore, restriction enzyme digestion on genomic DNA facilitated the separation of tandem copies of rDNA, largely reducing sample viscosity and enhancing template accessibility for downstream analyses [36]. Droplet generation using the QX200 droplet generator (Bio-Rad, Feldkirchen, Germany) was performed according to the manufacturer’s protocol, followed by an endpoint PCR. Amplifications were carried out with an initial denaturation at 95 °C for 10 min, 40 cycles of 96 °C for 30 s, 40 cycles of 54 °C (for humans) and 57 °C (for mice) for 56 s, and a final step at 98 °C for 10 min. A ramp rate of 2 °C/s was used. Subsequently, a QX200 droplet reader (Bio-Rad) was used to detect signals from individual droplets, and the data analyses were executed using QX Manager Standard Edition 2.0 software (Bio-Rad).

### 4.3. Deep Bisulfite Sequencing

DBS primers (Appendix A) were designed for the human rDNA UCE/CP region and the ETS as well as for the mouse rDNA CP, SP, and 28S rDNA. In human rDNA, both target regions contained an A/G variant [12,37]. For the UCE/CP variant (GRCh37; chr13: 999,905), >95% of reads represented the major G allele, and for the ETS > 80% the major A allele. To avoid an effect of genetic variation on our methylation results, in human sperm only reads representing the major variant were used for downstream analyses.

First-round PCR reactions were performed in 50 µL volumes, comprising 5 µL of 10x PCR buffer with MgCl_2_, 1 µL of PCR-grade nucleotide mixture (10 mM), 2.5 µL of forward and reverse primers (10 pmol/mL each), 0.4 µL of FastStart Taq DNA polymerase (5 U/µL), 2 µL of bisulfite-converted genomic DNA (~50 ng), and 36.6 µL of ddH_2_O. Additionally, completely unmethylated (0%) and completely methylated (100%) DNA (Qiagen) served as controls for assessing the reliability of methylation measurements for each DBS amplicon. PCR products were purified using Agencourt AMPure XP beads (Beckmann Coulter, Krefeld, Germany), quantified with a Qubit dsDNA BR Assay system kit (Invitrogen, Karlsruhe, Germany), and diluted to a final concentration of 0.2 ng/µL. In the subsequent PCRs, samples from different assays were pooled together and barcoded with multiple identifiers (MIDs). NEBNext Multiplex Oligos for Illumina (Dual Index Primers Set 1 and 2) were employed for the final PCR. Touch-down PCR thermocycler conditions were optimized to ensure uniform amplification of PCR templates of varying sizes. The purified and quantified PCR pools were diluted to a concentration of 4 nM, and 3 µL of this dilution from each of the MIDs was combined into a single final pool for next-generation sequencing (NGS).

NGS was performed using the MiSeq platform (Illumina, San Diego, CA, USA) and a Reagent Kit V2 (500 cycles) cartridge (Illumina) according to the manufacturer’s instructions. Sequencing involved 250 bp paired-end reads. Sequencing reads were processed by an Illumina Genome Analyzer. Further analysis of FASTQ files was conducted using Amplikyzer2 software (https://www.rahmannlab.de/software/amplikyzer, accessed on 1 April 2025) [38], which offers detailed nucleotide-level analysis and calculates methylation rates (per amplicon) at both single-nucleotide and regional levels. Initially, all sequences were aligned to the reference genomic sequence of each amplicon using default settings. For the subsequent extraction of reads and CpG-wise methylation status, only reads with an overall bisulfite conversion rate exceeding 95% were considered, and further downstream processing of Amplikyzer output files was performed.

### 4.4. Statistical Analysis

Both descriptive and inferential statistical analyses were conducted using IBM SPSS software version 28 and R version 3.6.3. Spearman’s correlations were employed to examine the relationships among variables such as CN, mean methylation, and age. In our human cohort, there was one sperm sample from a 72-year old donor. Since without this age outlier the correlations remained virtually the same, we kept this sample in our data set. Group comparisons, based on data distribution, were conducted using non-parametric Mann–Whitney U tests. The number of rDNA copies within a given methylation range (i.e., active copies with 0–10% methylation) was calculated by multiplying the absolute CN (from ddPCR) with the percentage of reads (from DBS) in the corresponding methylation bin (i.e., 0–10%). Throughout the analyses, a *p* value of less than 0.05 was considered statistically significant.

## Figures and Tables

**Figure 1 ijms-26-04197-f001:**
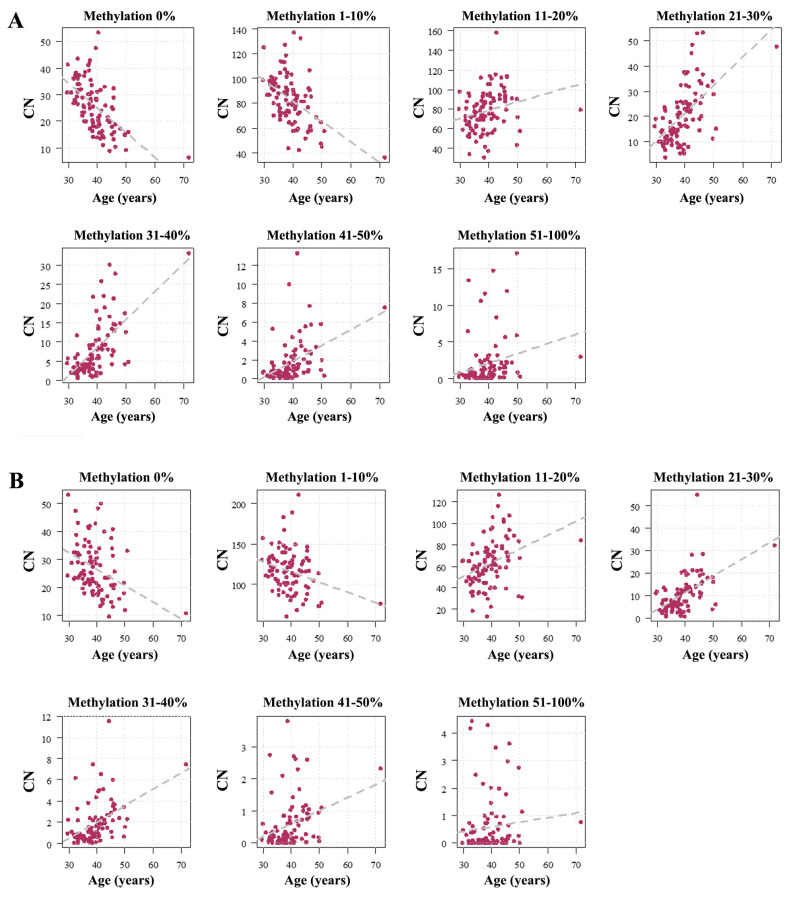
Age-related methylation changes of the rDNA UCE/CP (**A**) and the ETS (**B**). The Y axis shows the number of rDNA TUs within a given methylation bin. The first bin represents completely unmethylated TUs. The following bins represent mean methylation (across the entire region) from 1–10%, 11–20%, 21–30%, 31–40%, 41–50%, and 51–100%. Each dot represents an individual sperm sample. Please note that for both the UCE/CP and the ETS the number of completely unmethylated (0%) TUs and slightly methylated (1–10%) TUs decreases with age, whereas in all other bins the number of methylated (11–100%) TUs is increasing. For the UCE/CP (**A**), the Spearman correlations in the analyzed bins are as follows: 0% (*ρ* = −0.64, *p* < 0.0001), 1–10% (*ρ* = −0.50, *p* < 0.0001), 11–20% (*ρ* = 0.33, *p* = 0.001), 21–30% (*ρ* = 0.61, *p* < 0.0001), 31–40% (*ρ* = 0.63, *p* < 0.0001), 41–50% (*ρ* = 0.53, *p* < 0.0001), and 51–100% (*ρ* = 0.41, *p* < 0.0001). For the ETS (**B**), the Spearman correlations are the following: 0% (*ρ* = −0.41, *p* < 0.0001), 1–10% (*ρ* = −0.29, *p* = 0.005), 11–20% (*ρ* = 0.43, *p* < 0.0001), 21–30% (*ρ* = 0.59, *p* < 0.0001), 31–40% (*ρ* = 0.52, *p* < 0.0001), 41–50% (*ρ* = 0.35, *p* = 0.0006), and 51–100% (*ρ* = 0.19, *p* = 0.07).

**Figure 2 ijms-26-04197-f002:**
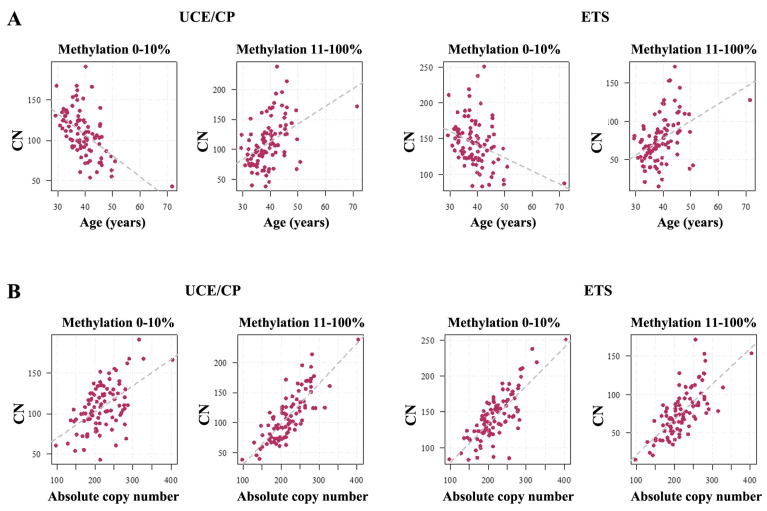
(**A**) The number of hypomethylated (0–10%) and hypermethylated (11–100%) rDNA TUs is negatively and positively correlated with age, respectively. For both the UCE/CP and the ETS regions, the number of hypomethylated (0–10%) copies is significantly decreasing (UCE/CP *ρ* = −0.56, *p* < 0.0001; ETS *ρ* = −0.35, *p* = 0.0005) with age, whereas the number of hypermethylated (11–100%) TUs is significantly increasing (UCE/CP *ρ* = 0.49, *p* < 0.0001; ETS *ρ* = 0.49, *p* < 0.0001). (**B**) The number of both hypomethylated (UCE/CP *ρ* = 0.48, *p* < 0.0001; ETS *ρ* = 0.72, *p* < 0.0001) and hypermethylated (UCE/CP *ρ* = 0.78, *p* < 0.0001; ETS *ρ* = 0.72, *p* < 0.0001) copies increases with absolute CN.

**Figure 3 ijms-26-04197-f003:**
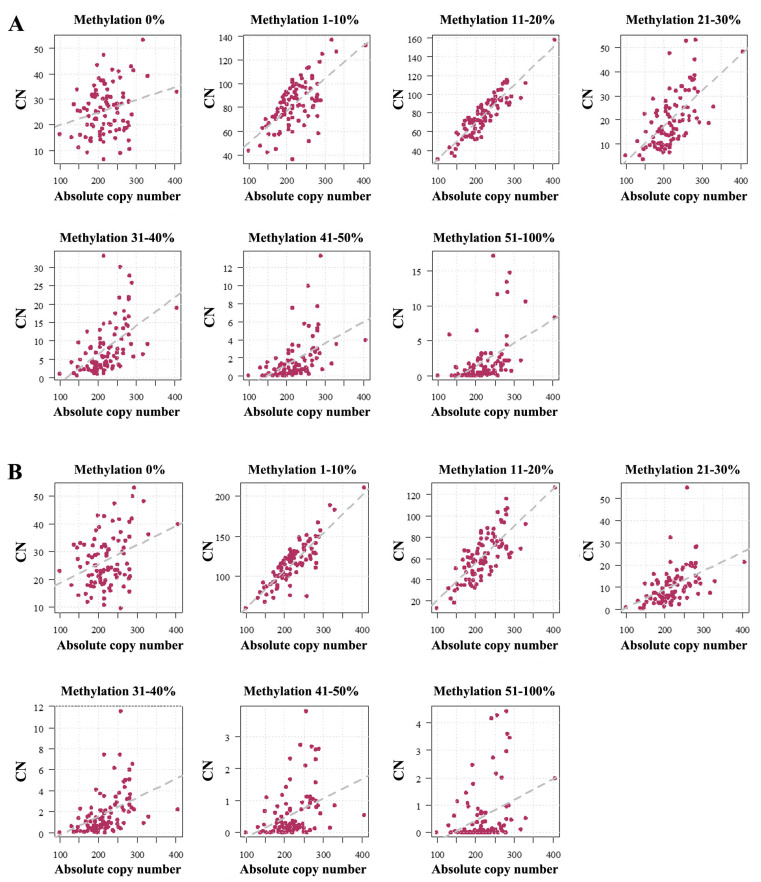
rDNA UCE/CP (**A**) and ETS (**B**) methylation depends on absolute copy number. Each dot represents an individual human sperm sample. The X axis shows the absolute number of rDNA TU copies. The Y axis presents the number of rDNA TUs within a given methylation bin (0%, 1–10%, 11–20%, 21–30%, 31–40%, 41–50%, and 51–100%). For both regions, the number of copies in all bins is increasing with absolute CN. For the UCE/CP (**A**), the Spearman correlations are as follows: 0% (*ρ* = 0.20, *p* = 0.05), 1–10% (*ρ* = 0.60, *p* < 0.0001), 11–20% (*ρ* = 0.87, *p* < 0.0001), 21–30% (*ρ* = 0.61, *p* < 0.0001), 31–40%% (*ρ* = 0.58, *p* < 0.0001), 41–50% (*ρ* = 0.59, *p* < 0.0001), and 51–100% (*ρ* = 0.54, *p* < 0.0001). For the ETS (**B**), the Spearman correlations are 0% (*ρ* = 0.28, *p* = 0.007), 1–10% (*ρ* = 0.81, *p* < 0.0001), 11–20% (*ρ* = 0.72, *p* < 0.0001), 21–30% (*ρ* = 0.56, *p* < 0.0001), 31–40% (*ρ* = 0.55, *p* < 0.0001), 41–50% (*ρ* = 0.44, *p* < 0.0001), and 51–100% (*ρ* = 0.33, *p* < 0.0001).

**Figure 4 ijms-26-04197-f004:**
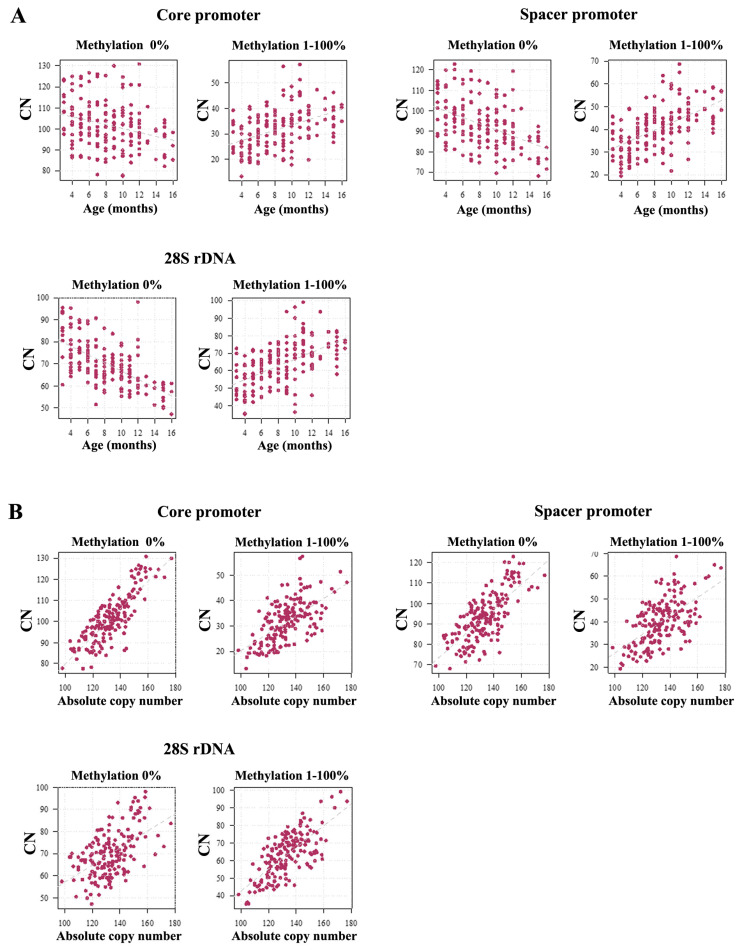
(**A**) The number of completely unmethylated (0%) and methylated (1–100%) rDNA TUs is inversely correlated with age. For the three target regions, the number of unmethylated copies is significantly decreasing (CP *ρ* = −0.28, *p* = 0.0002; SP *ρ* = −0.44, *p* < 0.0001; 28S *ρ* = −0.63, *p* < 0.0001) with age. In contrast, the number of methylated (1–100%) copies is significantly increasing (CP *ρ* = 0.45, *p* < 0.0001; SP *ρ* = 0.59, *p* < 0.0001; 28S *ρ* = 0.58, *p* < 0.0001). (**B**) The number of both hypomethylated (CP *ρ* = 0.80, *p* < 0.0001; SP *ρ* = 0.70, *p* < 0.0001; 28S *ρ* = 0.49, *p* < 0.0001) and methylated (CP *ρ* = 0.60, *p* < 0.0001; SP *ρ* = 0.56, *p* < 0.0001; 28S *ρ* = 0.67, *p* < 0.0001) copies increases with absolute CN.

## Data Availability

All data are contained in the manuscript and its Appendix A.

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
