# Peer review of "rDNA Copy Number Variation and Methylation in Human and Mouse Sperm"

_ijms, 2025, doi:10.3390/ijms26094197_

Round 1

Reviewer 1 Report

Comments and Suggestions for Authors

In the current manuscript, the authors determine the rDNA copy number (CN) using droplet digital PCR and deep bisulfite sequencing. Little is known about the CN variation in sperm. The authors performed the methylation status. The study is interesting. The manuscript is well-written, and the results support the conclusions. I have the following minor comments:

  • How did the authors compare the sperm parameters of the mouse and human?
  • The authors should provide a graphical abstract of their findings.

Author Response

COMMENT 1:  How did the authors compare the sperm parameters of the mouse and human?

RESPONSE 1:   Although the human sperm doners were attending a fertility center, they had normal semen parameters according to the 5th edition of the WHO laboratory manual, which is generally considered as an indicator for male fertility potential. Although we do not have semen parameters for the mouse sperm samples, they came from fertile healthy animals. In the present study we do not associate rDNA copy number and methylation with semen parameters or fertility status. Our aim was to study absolute and active copy number variation in human and mouse.

COMMENT 2:   The authors should provide a graphical abstract of their findings.

RESPONSE 2:   A graphical abstract has been provided.

Thank you very much for your efforts regarding our manuscript.

Reviewer 2 Report

Comments and Suggestions for Authors

‘rDNA Copy Number Variation and Methylation in Human and Mouse Sperm’ is an original article in which authors investigated whether ribosomal DNA copy number variation differ between mouse and human. They also investigated whether methylation patterns between mouse and human sperm differ. By investigating this, authors tried to gain some new insights into male germline programming and its consequences in embryo development and later in life.

The work fit the journal and special issue scope. The manuscript is clear, relevant for the field and presented in a well-structured manner. The cited references are mostly recent publications (within the last 5 years) and relevant. Cited references do not include an excessive number of self-citations. The manuscript is scientifically sound and the experimental design is appropriate to test the hypothesis although the hypothesis is not well defined. The manuscript’s results are reproducible based on the details given in the methods section. Figures/tables/images/schemes are appropriate. They properly show the data but they are not always easy to interpret and understand. Some additional clarification would be helpful. The data are interpreted appropriately and consistently throughout the manuscript. The conclusions are consistent with the evidence and arguments presented. The ethics statements and data availability statements are adequate.

Although the results provide an advancement of the current knowledge and although the scientific question (hypothesis) is original, it is not well-defined, neither in abstract nor in introduction section. Please rewrite this so that it is clear what was the scientific hypothesis of this work.

The study is technically sound but not completely correctly designed (as stated earlier). The analyses are performed with the highest technical standards.  The methods, tools, software, and reagents are described with sufficient details to allow another researcher to reproduce the results. The raw data are available.

In the discussion section, authors did not clearly stated what are the limitations of their investigation. Please add this.

I find the results and conclusions interesting for the readership of the journal and this particular special issue. Although the results are interested, it will be of interest only to a limited number of people. Nevertheless, I do think that this work advances the current knowledge about the sperm epigenome and that this paper should be published after the minor corrections are being done.

Author Response

COMMENT 1:   Although the results provide an advancement of the current knowledge and although the scientific question (hypothesis) is original, it is not well-defined, neither in abstract nor in introduction section. Please rewrite this so that it is clear what was the scientific hypothesis of this work.

RESPONSE 1:  Due to the word maximum it is difficult to expand on our working hypothesis in the abstract. However, we have clarified the aim of our study at the end of the Introduction section. It is well known that absolute rDNA CN in somatic tissues can considerably vary between individuals and also between species. We hypothesize that redundant copies are inactivated by rDNA methylation. Consequently, active CN should be less variable. Recently, we have developed a combination of droplet digital PCR and deep bisulfite sequencing, which allows us to count not only the absolute rDNA CN, but also the number of presumably active copies with a hypomethylated promoter region with an unprecedented accuracy. To find out whether haploid germ cells exhibit the same CN variation and age-dependent gain of methylation as somatic cells (blood), we have determined the number of rDNA TU with 0%, 10%, 20%, etc. methylation in human and mouse sperm. Despite enormous species differences in absolute rDNA CN, the number of hypomethylated copies and the paternal age effect were comparable between humans and mice. An evolutionarily conserved rDNA methylation clock appears to operate in similar ways in the mammalian soma and germ line.

COMMENT 2:  The study is technically sound but not completely correctly designed (as stated earlier). The analyses are performed with the highest technical standards. The methods, tools, software, and reagents are described with sufficient details to allow another researcher to reproduce the results. The raw data are available.

RESPONSE 2:   Our study was designed to determine absolute and active rDNA copy number and their variation in human and mouse. Our study provides the best estimate for rDNA CN so far and can serve as a reference. However, we can only speculate about functional implications of copy number differences between individuals or between species.

COMMENT 3:  In the discussion section, authors did not clearly stated what are the limitations of their investigation. Please add this.

RESPONSE 3:  :  We have re-structered the discussion and added a Limitations section at the end. One important point are the species differences in reproduction, embryogenesis and lifespan, which make it difficult to compare absolute and active copy numbers between human and mouse sperm and their functional implications. “When it comes to studying ageing …, mice are not just small humans.” Overall the paternal age effect on rDNA methylation has been conserved, leading to an decrease of presumably active sperm DNA copies with age. This may contribute to the effect of paternal age on offspring development. However, this largely conjectural as so far there is no evidence of a functional outcome in the post-fertilization early embryo.     Another limitation is that we do not really know to which extent unmethylated (0%) or lowly (1-10%) methylated rDNA copies are transcribed. It is difficult to prove by functional experiments, which methylated CpG density is required for epigenetic silencing and there may not be a sharp threshold anyways. Moreover there may be species differences. Our data suggest that lowly (1-10%) methylated rDNA copies are active in humans but not in mice.

Thank you very much for your efforts regarding our manuscript.